# Chemical, physical, and functional properties of Thai indigenous brown rice flours

**David Oppong, Worawan Panpipat[ID], Manat Chaijan[ID]***

Food Technology and Innovation Research Center of Excellence, School of Agricultural Technology and Food Industry, Walailak University, Thasala, Nakhon Si Thammarat, Thailand

* cmanat@wu.ac.th

## Abstract

Thai indigenous brown rice flours from Nakhon Si Thammarat, Thailand, namely Khai Mod Rin (KMRF) and Noui Khuea (NKRF), were assessed for quality aspects in comparison with brown Jasmine rice flour (JMRF) and commercial rice flour (CMRF) from Chai Nat 1 variety. All the rice flours had different chemical composition, physical characteristic, and techno-functionality. The KMRF, NKRF, and JMRF were classified as a low amylose type (19.56–21.25% dw). All rice flours had low total extractable phenolic content (0.1–0.3 mg GAE/g dw) with some DPPH$^\bullet$ scavenging activity (38.87–46.77%). The variations in the bulk density (1.36–1.83 g/cm$^3$), water absorption capacity (0.71–1.17 g/g), solubility (6.93–13.67%), oil absorption capacity (1.39–2.49 g/g), and swelling power (5.71–6.84 g/g) were noticeable. The least gelation concentration ranged from 4.0 to 8.0% where KMRF was easier to form gel than JMRF, and NKRF/CMRF. The foam capacity of the flours was relatively low (1.30–2.60%). The pasting properties differed among rice flours and the lowest pasting temperature was observed in CMRF. Overall, the chemical, physical, functional, and pasting qualities of flours were substantially influenced by rice variety. The findings offered fundamental information on Thai indigenous rice flour that can be used in food preparations for specific uses.

## Introduction

As one of the pivotal cereal grains in the world, rice (*Oryza sativa* L.) is a good source of essential nutrients, especially carbohydrate and protein [1]. The global production of rice is estimated to be doubly increased by 2050 due to the increasing consumer demand [2]. Rice is mostly consumed as intact kernels, but rice flour can be used for several food preparations such as traditional foods, noodles, baked goods, extruded products, and novel products (e.g. gluten free based foods, infant's foods, and snacks) [1, 3]. It has been reported that the physico-chemical properties of rice flour significantly affected the qualities of those products. Commercial rice flour (CMRF) is primarily produced from Chai Nat 1 white rice to meet appreciations of consumers [1, 3], but the flour from the polished grain contains lower nutrients and bioactive compounds due to the elimination of the bran upon polishing/milling [4]. In Thailand, waxy and high amylose rice varieties are regularly used for CMRF production [5]. Owing to

**Data Availability Statement:** All relevant data are within the manuscript.

**Funding:** This research was funded by Walailak University, Thailand through the Ph.D. Scholarship

for Outstanding International Students, grant number [MOE 57 19 00/144/2562]. This research was also financially supported by the new strategic research project (P2P), Walailak University, Thailand. The funders had no role in study design, data collection and analysis, decision to publish, or preparation of the manuscript.

**Competing interests:** The authors have declared that no competing interests exist.

the consumers' demands, indigenous rice flour has been paid more attention as an alternative raw material for several food products.

Compositional variations in terms of physical, chemical, thermal, and pasting properties were found among rice varieties, depending on the genotype, agronomic and cultivation conditions, environmental factors, storage conditions, and processing parameters [4, 6]. The quality of rice does not only include the physical appearance, rather it encompasses the chemical, functional, and thermal properties [6]. In Thailand, different indigenous rice varieties have been grown. In Southern Thailand, more than 4,000 local rice varieties have been recorded [7, 8]. Especially in Nakhon Si Thammarat, non-glutinous domestic rices such as Khai Mod Rin and Noui Khuea have been widely cultivated following the production practices for organic Thai rice. Recently, the health benefits for the consumption of brown rice were intensively reported [4]. Therefore, in this research, brown Khai Mod Rin rice flour (KMRF) and brown Noui Khuea rice flour (NKRF), which are mainly cultivated in Nakhon Si Thammarat, Thailand, were investigated for their basic chemical composition, physical property, and techno-functionality in comparison with organic brown Jasmine rice flour (JMRF) and CMRF.

## Material and methods

### Chemicals

All chemicals used for analyses such as DPPH (2,2-Diphenyl-1-picrylhydrazyl), acetone, methanol, and gallic acid (GA) were obtained from Sigma-Aldrich Corp. (St. Louis, MO, USA).

### Raw materials

Two domestic Southern Thai brown rices (*Oryza sativa* L., varieties Khai Mod Rin and Noui Khuea) and brown Jasmine rice were obtained from an organic farm in Phra Phrom, Nakhon Si Thammarat, Thailand (8˚17'23.4"N, 99˚58'48.2"E, altitude of 9 m), in March 2020. To prepare the flour, brown rices were ground for 5 min using a grinder (MK 5087M Panasonic Food Processor, Selangor Darul Ehsan, Malaysia) and passed through a 100-mesh sieve. CMRF, from Chai Nat 1 variety, was obtained from Cho Heng Rice Vermicelli Factory Co., Ltd., Nakhon Pathom, Thailand (13˚43'38.1"N, 100˚14'42.3"E, altitude of 10 m), in March 2020. The flours were packed in polythene bags and kept at room temperature (28–30˚C) until further analysis. The storage time was less than a month. Three different lots of flour were prepared to get triplications for all analyses. The contents for all chemical compositions were reported on a dry weight (dw) basis.

### Proximate composition

The standard methods of AOAC [9] were used for proximate composition analysis including moisture (AOAC method number 950.46), crude protein (AOAC method number 928.08, a conversion factor = 5.95), ash (AOAC method number 920.153), fiber (AOAC method number 962.09), fat (AOAC method number 963.15), and carbohydrate (calculated by difference).

### Amylose content

Amylose content was determined according to Kraithong et al. [10]. The sample (100 mg) was mixed with 95% (v/v) ethanol (1 mL) and 2 M NaOH (9 mL). The mixture was brought to 100 mL with distilled water and then 0.2% (w/v) iodine solution (2 mL) was added. Thereafter, the absorbance was read at 620 nm (Shimadzu UV-2100 spectrophotometer, Shimadzu Scientific Instruments Inc., Columbia, MD, USA). A calibration curve was created using standard amylose obtained from potato starch to quantify the amylose content.

## Total extractable phenolic content (TPC) and DPPH$^\bullet$ scavenging activity

The method of Sungpud et al. [11] was used for TPC extraction. The flour samples (10 g) were extracted with 80% (v/v) ethanol (100 mL) at 40°C in the Lib-300M incubator (Labtech, Korea) for 24 h under continuous magnetic stirring (300 rpm). Thereafter, the mixtures were centrifuged at 5,000 rpm for 10 min at room temperature (RC-5B plus centrifuge, Sorvall, Norwalk, CT, USA) and the supernatants were collected. After filtration (Whatman No.1), the filtrates were analyzed for TPC. One hundred µL of ethanolic flour extract were mixed with 2.0 mL Folin-Ciocalteu reagent (previously diluted to 10-fold with deionized water) and well mixed. After standing for 5 min, 15% sodium carbonate solution (1.0 mL) was added. The correspondence solution was kept in the dark for 60 min. The absorbance was read at 765 nm using a UV-vis spectrophotometer (Shimadzu, MD, USA). The TPC was expressed as mg gallic acid equivalent (GAE)/g sample.

The DPPH$^\bullet$ scavenging effect was analyzed according to Limsuwanmanee et al. [12]. Ethanolic flour extract (1 mL; 0.1 mg/mL TPC) was mixed with 0.2 mM methanolic DPPH$^\bullet$ solution (1 mL). After incubation in the dark at room temperature (30 min), the absorbance was measured at 517 nm against blank. A control was prepared using methanol instead of the sample. DPPH$^\bullet$ inhibition was acquired by the following formula:

$$\text{DPPH}^\bullet \text{ inhibition (\%)} = [(A_0 - A_1)/A_0] \times 100 \tag{1}$$

where $A_0$ = absorbance of the control and $A_1$ = absorbance of the sample.

## Fourier transform infrared (FTIR) spectroscopy

The FTIR spectroscopy is a vibrational spectroscopic technique that can be used to characterize the substances by identifying their functional groups presented [13]. FTIR spectra (500–4,000 cm$^{-1}$ with the resolution of 4 cm$^{-1}$ at the average of 16 scans) of the rice flours were obtained using a horizontal Attenuated Total Reflectance (ATR) Trough plate crystal cell (45° ZnSe; 80 mm long, 10 mm wide and 4 mm thick) (Pike Technology, Inc., Madison, WI, USA) equipped with a Bruker Model Vector 33 FTIR spectrometer (Bruker Co., Ettlingen, Germany) at room temperature. Analysis of spectral data was carried out using the OPUS 3.0 data collection software program.

## Color

The rice flour color was determined using a Hunterlab colorimeter with 10 standard observers and illuminant D65 (Hunter Assoc. Laboratory; VA, USA). The $L^*$, $a^*$, and $b^*$ values were recorded.

## Bulk density

Fifty grams of flour was taken to a measuring cylinder (100 mL) and tapped carefully. After reading the volume, the bulk density was estimated from the ratio of mass (g) to volume (mL) [14].

$$\text{Bulk density (g/mL)} = \frac{\text{Weight of flour (g)}}{\text{Volume of flour after settled (mL)}} \tag{2}$$

## Water absorption capacity (WAC) and solubility

The WAC and solubility were determined using the method of Kraithong et al. [10]. One gram of flour was suspended in 10 mL of distilled water and mixed with a vortex mixer for 1 min. The suspensions were heated in a water bath at 30°C for 30 min with gentle stirring and

then centrifuged at 1,500 ×g for 10 min (RC-5B plus centrifuge). The supernatant was carefully poured into an aluminum moisture can before being dried at 105°C overnight. The sediments were collected and weighed. The WAC and solubility were calculated using the following formulas:

$$\text{WAC (g/g)} = \frac{\text{Weight of wet sediment (g)}}{\text{Dry weight of flour (g)}} \tag{3}$$

$$\text{Solubility (\%)} = \frac{\text{Weight of dried supernatant (g)}}{\text{Dry weight of flour (g)}} \times 100 \tag{4}$$

### Oil absorption capacity (OAC)

The OAC of the rice flour was measured according to Kraithong et al. [10]. One gram of flour was mixed with 10 mL of soybean oil for 1 min. After standing at room temperature (30 min), the centrifugation was applied at 1,500 ×g for 10 min (RC-5B plus centrifuge). Thereafter, the surplus oil was decanted while the residue (weight of oil absorbed) was weighed. The calculation of OAC was as follows:

$$\text{OAC (g/g)} = \frac{\text{Weight of oil absorbed (g)}}{\text{Weight of sample (g)}} \tag{5}$$

### Swelling power

The swelling power was measured according to Appiah et al. [15]. The flour sample (1 g) was mixed with distilled water (30 mL). After heating (85°C/30 min) in a W350 Memmert water bath (Schwabach, Germany), sample was cooled to room temperature and centrifuged at 1,500 ×g for 20 min (RC-5B plus centrifuge). The swelling power was estimated as the weight of the paste per weight of the dry sample.

$$\text{Swelling power (g/g)} = \frac{\text{Weight of the paste (g)}}{\text{Weight of dry sample (g)}} \tag{6}$$

### Least gelation concentration (LGC)

The LGC was measured according to Appiah et al. [15]. Aqueous suspensions of 2, 4, 6, 8, up to 20% (w/v) flour (5 mL) were heated in boiling water for 1 h. After cooling down in ice bath and standing at 4°C for 2 h, the test tube was inverted. The LGC was regarded as the concentration at which the inverted sample did not slip down the side of the test tube.

### Foaming capacity (FC)

The flour sample (3 g) was mixed with distilled water (100 mL) at room temperature and homogenized for 5 min at 13,400 rpm using an IKA® homogenizer (Model T25 digital Ultra-Turrax®, Staufen, Germany). The increase in volume of the foam at 30 s after whipping against the original volume was expressed as FC [16].

$$\text{FC (\%)} = \frac{\text{Volume after whipping (mL)}}{\text{Original voloume (mL)}} \times 100 \tag{7}$$

### Pasting properties

Pasting characteristics were analyzed using the Rapid Visco Analyzer (RVA 4500, Perten Instruments, Sweden). In a canister, the rice flour sample (3 g) was inserted, and then 25 mL of distilled water was added (14% moisture basis). The RVA profile was recorded under

specific conditions. The temperature was held at 50˚C for 1 min and then raised up to 95˚C in 3.8 min (held for 2.5 min). Consequently, it was cooled to 50˚C within 3.8 min and held for 1.4 min. The pasting parameters e.g. pasting temperature, peak viscosity, breakdown, final viscosity, and setback were measured according to the method of Kraithong et al. [10].

### Statistical analysis

A completely randomized design was used for experimental design with a single factor of four rice flours and the experiments were performed in triplicate. Data were subjected to one-way analysis of variance (ANOVA). Duncan's multiple-range test was used to analyze significant differences ($p < 0.05$) among samples, using the SPSS program (Version 23.0, SPSS Inc., Chicago, IL, USA).

## Results

### Proximate composition and amylose content

The proximate compositions of Thai indigenous brown rice flour, namely KMRF and NKRF, in comparison with JMRF and CMRF are presented in Table 1. Generally, the brown rice flours had higher contents of protein, ash, fiber, and fat than CMRF, whereas CMRF had a higher carbohydrate content ($p < 0.05$). The moisture contents of the rice flours were in the range of 4.25–5.06% and the lowest content was found in CMRF ($p < 0.05$). The flours had protein ranging between 5.01% and 8.14%. The highest protein content was found in NKRF ($p < 0.05$), followed by KFRF/JMRF and CMRF. The ash contents of rice flours were in the range of 0.38–2.22%. No significant difference in ash content was found among the brown rice varieties (2.0–2.2%) ($p > 0.05$). The crude fiber content of the flours ranged from 0.49% to 2.85%. JMRF had the highest crude fiber (2.85%), followed by KMRF (2.54%), NKRF (2.06%), and CMRF (0.49%). The fat content of the rice flours ranged from 0.77% to 2.0%. NKRF had the highest fat (2.0%), whereas the lowest value was found in CMRF (0.77%). In the present experiment, carbohydrate was found to be high in all samples (>85%). CMRF had a higher carbohydrate content (93.3%) than brown rice flours (85.79–87.82%).

According to Table 1, there is no significant variation in amylose content for the three brown rice flours ($p > 0.05$). However, the CMRF was markedly different from them ($p < 0.05$). The amylose contents in the rice flours were 19.56–21.25% (Table 1).

**Table 1. Chemical compositions of Thai indigenous brown rice flour, namely brown Khai Mod Rin rice flour (KMRF) and brown Noui Khuea rice flour (NKRF), in comparison with brown Jasmine rice flour (JMRF) and commercial rice flour (CMRF).**

| Chemical composition | Rice flour | | | |
|---|---|---|---|---|
| | **KMRF** | **NKRF** | **JMRF** | **CMRF** |
| Moisture (% ww) | 5.00 ±0.20a | 5.06 ±0.12a | 4.93 ±0.11a | 4.25 ±0.49b |
| Protein (% dw) | 6.35 ±0.15b | 8.14 ±0.63a | 7.01 ±0.75b | 5.01 ±0.26c |
| Ash (% dw) | 2.20 ±0.01a | 1.99 ±0.05a | 2.22 ±0.25a | 0.38 ± 0.00b |
| Fiber (% dw) | 2.54 ±0.00b | 2.06 ±0.02c | 2.85 ±0.02a | 0.49 ± 0.02d |
| Fat (% dw) | 1.08 ±0.15bc | 2.00 ±0.10a | 1.39 ±0.23b | 0.77 ±0.17c |
| Carbohydrate (% dw) | 87.82 ±0.05b | 85.79 ±0.74c | 86.54 ±0.52c | 93.32±0.54a |
| Amylose (% dw) | 19.56 ±0.23b | 19.60 ±0.01b | 19.84 ±0.08b | 21.25±0.23a |
| Total extractable phenolic content (mg GAE/g dw) | 0.32 ±0.02a | 0.34 ±0.01a | 0.25 ±0.01b | 0.12 ±0.01c |
| DPPH radical scavenging activity (%) | 38.87 ±0.29b | 37.40 ±0.97bc | 35.11 ±0.72c | 46.77±1.69a |

Values are means ± standard deviation from triplicate determinations. Different letters are significantly different along the rows ($p < 0.05$).

GAE = gallic acid equivalent, ww = wet weight, dw = dry weight

## TPC, DPPH$^{\bullet}$ scavenging activity, and FTIR spectra

The TPC of the rice flours are given in Table 1. The TPC in the brown rice flours was higher than the CMRF ($p < 0.05$). The highest TPC in the brown rice flours was owned by NKRF and KMRF (~0.3 mg GAE/g), followed by JMRF (0.2 mg GAE/g), and CMRF (0.1 mg GAE/g). The antioxidant activity of ethanolic extracts of rice flours was determined by DPPH$^{\bullet}$ assay (Table 1). Inactivation of DPPH$^{\bullet}$ was found in all of the rice flour extracts examined, ranging from 38.87 to 46.77%. The CMRF extract showed higher antioxidant capacity than those of the brown rice flours. The FTIR spectra for rice flours are depicted in Fig 1. All the samples showed similar spectra within a region of 500–4000 cm$^{-1}$ with different peak intensities.

## Color and bulk density

The color attributes ($L^*$, $a^*$, and $b^*$) of the rice flours were remarkably different ($p < 0.05$) (Table 2). $L^*$ values, which express the lightness, were in the range of 75.85–95.20 with the highest $L^*$ value coming from CMRF. Generally, the $L^*$ value of CMRF was higher than NKRF, KMRF, and JMRF, respectively ($p < 0.05$). Higher values of $a^*$ and $b^*$ with lower $L^*$ value were found in all brown rice flours ($p < 0.05$), while a negative $a^*$ value, a low $b^*$ value and the highest $L^*$ value was noticeable in CMRF ($p < 0.05$). The rice flours had bulk densities ranging between 1.36 g/mL to 1.83 g/mL (Table 2). The mean bulk density was ranged in the order of CMRF = NKRF ≤ JMRF ≤ KMRF.

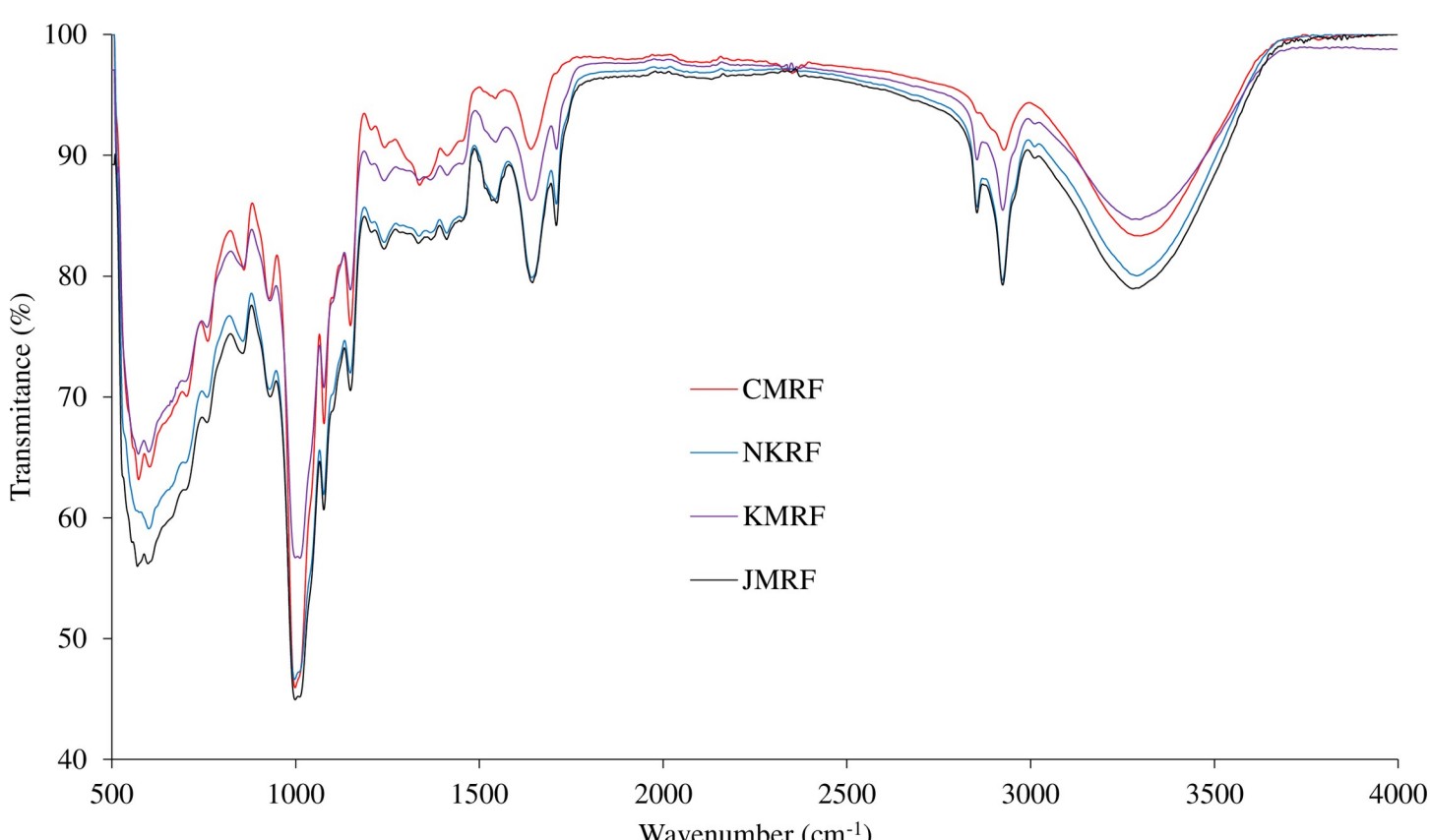

**Fig 1. FTIR spectra of Thai indigenous organic brown rice flour, namely brown Khai Mod Rin rice flour (KMRF) and brown Noui Khuea rice flour (NKRF), in comparison with brown Jasmine rice flour (JMRF) and commercial rice flour (CMRF).**

**Table 2. Physical and functional properties of Thai indigenous brown rice flour, namely brown Khai Mod Rin rice flour (KMRF) and brown Noui Khuea rice flour (NKRF), in comparison with brown Jasmine rice flour (JMRF) and commercial rice flour (CMRF).**

| Parameters | Rice flour | | | |
|---|---|---|---|---|
| | **KMRF** | **NKRF** | **JMRF** | **CMRF** |
| Color | | | | |
| $L^*$ | 77.96±0.04c | 79.74 ±0.02b | 75.85 ±0.03d | 95.20 ±0.06a |
| $a^*$ | 2.24 ±0.02b | 1.63 ±0.07c | 2.64 ±0.06a | -0.11 ±0.00d |
| $b^*$ | 13.64 ±0.15c | 14.35 ±0.05b | 15.73 ±0.18a | 3.14 ±0.11d |
| Bulk density (g/mL) | 1.36 ±0.05b | 1.78 ± 0.00a | 1.69 ± 0.28ab | 1.83 ±0.23a |
| Water absorption capacity (WAC; g/g) | 1.16 ±0.33a | 1.06 ±0.06ab | 1.17 ±0.16a | 0.71 ±0.06b |
| Solubility (%) | 13.67±5.09a | 10.83±1.04ab | 13.40 ± 1.96a | 6.93 ±2.18b |
| Oil absorption capacity (OAC; g/g) | 2.31±0.41a | 2.37 ±0.65a | 1.39 ±0.98a | 2.49 ±0.40a |
| Swelling power (g/g) | 6.84±0.45a | 6.49 ±0.39ab | 6.51 ± 0.76ab | 5.71±0.25b |
| Least gelation concentration (LGC; %) | 4.00 ± 1.00c | 8.00 ± 1.00a | 6.00±1.00b | 8.00±1.93a |
| Foam capacity (FC; %) | 2.61±1.13a | 1.50±0.10b | 2.00±0.00ab | 1.30±0.17b |

Values are means ± standard deviation from triplicate determinations. Different letters are significantly different along the rows ($p < 0.05$).

WAC, solubility, OAC, swelling power, LGC, and FC

In this study, functional properties of rice flour were tested for WAC, solubility, OAC, swelling power, LGC, FC, and pasting characteristics. WAC of the rice flours ranged from 0.71 g/g to 1.17 g/ g as shown in Table 2. The lowest WAC was found in CMRF (0.71 g/g), followed by NKRF and KMRF/JMRF. The solubility ranged from 6.93% to 13.67% (Table 2). There were no significant differences in terms of OAC among the flours ($p > 0.05$). The OAC of the rice flours ranged from 1.39 g/g to 2.49 g/g (Table 2). In addition, for all flours, the OAC was higher than the WAC (Table 2). The flours had swelling power values ranging between 5.71 g/g and 6.84 g/g (Table 2). The LGC values ranged from 4.0% to 8.0% (Table 2). The FC of the flours was relatively low and the values ranged from 1.30% to 2.6% (Table 2). With regards to the FC, the value of KMRF ≥ JMRF ≥ NKRF = CMRF.

## Pasting properties

The pasting properties of the rice flours are given in Table 3. The pasting temperature of the rice flours varied between 89°C and 91°C (Table 3). All brown rice flours had a higher pasting temperature than CMRF ($p < 0.05$). The peak viscosity of the rice flours under study ranged from 847 BU to 2,250 BU, as shown in Table 3. The result indicated that the CMRF had

**Table 3. Pasting of Thai indigenous brown rice flour, namely brown Khai Mod Rin rice flour (KMRF) and brown Noui Khuea rice flour (NKRF), in comparison with brown Jasmine rice flour (JMRF) and commercial rice flour (CMRF).**

| Parameters | Rice flour | | | |
|---|---|---|---|---|
| | **KMRF** | **NKRF** | **JMRF** | **CMRF** |
| Pasting temperature (°C) | 91±0a | 91±0a | 91±0a | 89±1b |
| Peak viscosity (BU) | 898±12c | 847±12d | 1,230±24b | 2,250±31a |
| Trough viscosity (BU) | 872±16c | 589±23d | 1,234±23b | 1,927±9a |
| Breakdown (BU) | 19 ±11c | 258±19b | -3±27c | 321±25a |
| Setback (BU) | 142±1b | -85±4d | 312±6a | 100±18c |
| Final viscosity (BU) | 1021±1c | 497±12d | 1545±1b | 2,028±24a |

Values are means ± standard deviation from triplicate determinations. Different letters are significantly different along the rows ($p < 0.05$).

significantly higher peak viscosity followed by JMRF, KMRF, and NKRF (Table 3). The trough viscosity of the flours ranged from 589 BU to 1,927 BU, as illustrated in Table 3. The highest setback value was found in JMRF (312 BU) and the highest final viscosity was found in CMRF (2,028 BU) ($p < 0.05$; Table 3).

## Discussion

Because to varietal differences, geographical location, and processing conditions, all of the rice flours had varied proximate compositions. The moisture contents of the rice flours in this study were similar to those reported by Kraithong et al. [10] for Thai pigmented and non-pigmented rice flour (5.47–9.87%), but they were lower than those reported previously for Thai jasmine red rice flour (13.3%) [17] and whole flour from Thai purple rice (11.57%) [18]. The moisture content of the flours in this investigation, however, was less than the critical moisture content of 13% [19]. The higher the moisture level of flour, the more likely it is to be spoiled by microorganisms. As a result, the flours in this study should have a good shelf life. The protein amount found in this study was comparable to that found in Thai purple rice flours (6.6–13.0%) [18], black glutinous rice flour (8.0%) [20], and black rice variety (8.0%) [21]. Generally, rice grain storage proteins are composed of albumins, globulins, glutelins and prolamins, which are soluble in water, salt solution, dilute acid or alkaline solutions, and aqueous alcohol, respectively [22]. The ash content of rice flours ranged from 0.38% to 2.22%. The ash content of 1.8% on brown rice flours reported by Islam et al. [21] was comparable to the values (2.0–2.2) found in our investigation. The results of this study were within the range of 0.43–2.34% reported by Devi et al. [19] on 92 rice varieties, including indigenous, improved, and fragrant types. The ash content reflected the mineral content in the sample. Thomas et al. [20] reported ash content of 0.90% in black rice and 0.39% in white rice. The CMRF was significantly lower than the brown rice flour in this investigation, indicating the same tendency. When compared to the CMRF, this revealed that brown rice flours could be key sources of minerals.

The crude fiber contents of the rice flours in this study (0.49–2.85%) were similar to those found in earlier studies. The fiber content of 1.23–1.56% was reported in brown rice flour [3, 23] whereas the content of 0.34% was reported in CMRF [24]. According to Oko and Onyekwere [24], the average content of fiber in well-milled rice flour was around 0.5–1.0%. The higher fiber content in brown rice flour was due to the presence of bran fraction. Dietary fiber has a number of health benefits, including lowering blood cholesterol and/or glucose levels, acting as a laxative, and lowering the risk of colon cancer, heart disease, and obesity [25]. The principal components of dietary fiber present in rice are arabinoxylans, β-glucans, cellulose, and hemicellulose [26, 27]. The number and quantity of these non-starch polysaccharides in rice, on the other hand, are determined by the rice cultivar, milling degree, and water solubility. Again, the higher fat content in brown rice flour was due to the presence of bran fraction. Here, the fat contents for the three brown rice varieties were among the values of brown rice varieties (0.2–3.86%) reported by Devi et al. [19] and Ye et al. [28]. The fat content of CMRF in this research was lower than Phitsanulok white rice (1.13%) [10]. Fat content had a substantial positive link with sensory total points of eating quality of rice, according to Ke-xin et al. [29]. In this study, carbohydrate was higher than 85%. The carbohydrate content of 80.35–91.33% has been reported in CMRF [23] whereas the value of 77.31% was found in brown rice flour [21]. A lower carbohydrate content in the brown rice varieties was attributed to higher contents of protein, fat, ash, and fiber.

Herein, the amylose contents in the rice flours were around 20–21%. Juliano [30] classified the rice based on the amylose content, namely waxy (0–5%), very low (5–12%), low (12–20%), intermediate (20–25%), and high (25–33%). This implied that the brown rice flours in this

study were classified as a low amylose type whereas the CMRF was an intermediate type. The results were in agreement with Ye et al. [28] who reported the amylose content of 10.4–26.5% in India rice flour. According to Falade and Christopher [31], low amylose rice flour provides moistness, softness, and chewiness to product textures.

Brown rice is high in phenolic compounds, according to several studies [32–35]. The phenolic compounds are collectively composed of phenoic acids, flavonoids, and anthocyanins/proanthocyanidins [32]. Several phenolic compounds such as gallic acid, protocatechuic acid, *p*-hydroxybezoic acid, vanillic acid, syringic acid, chlorogenic acid, caffeic acid, *p*-coumaric acid, sinapic acid, ferulic acid, cinnamic acid, ellagic acid, luteolin, apigenin, tricin, quercetin, kaempferol, isorhamnetin, myricetin, etc have been identified in brown rice [32–35]. The TPC obtained from this study was within those reported by Ponjanta et al. [5] who found TPC around 0.3–2.4 mg GAE/g in Thai rice flours. The highest TPC in the brown rice flours was found in NKRF/KMRF, followed by JMRF, and CMRF. This implied that indigenous rice flour is a better source of phenolic compounds. The variation of TPC in brown rice varieties was governed by genotype, cultivation techniques, and environmental factors [36]. Grain phenolic compounds were eliminated during rice milling and flour preparation, as indicated by a low TPC in CMRF. In comparison to unmilled rice, Sapna et al. [37] found that the TPC of milled rice retained roughly 73–89% of the overall TPC.

However, the extractability of the phenolic compounds may also influence the TPC of the flours. In rice, phenolics can be found in three different forms including free, soluble-conjugated, and bound forms [33]. Free phenolics are the most readily available for absorption in the small intestine [33]. The acidified solvent can be used to enhance the extraction of bound phenolics (e.g. anthocyanins) from pigmented rice [35]. However, in this study, the non-pigmented rice were used. So, the extraction with 80% ethanol, one of the common solvents used for TPC isolation, at 40˚C for 24 h under continuous stirring was reasonable for TPC recovery.

For the antioxidant activity of ethanolic extracts of rice flours indicated by DPPH• scavenging activity, the CMRF extract showed higher antioxidant capacity than those of the brown rice flours. Maisuthisakul and Changchub [38] reported that white rice varieties had relatively higher antioxidant power than red rice. In their study using ethanol with shaking method, the DPPH• inhibitions were ranged from 56.21% to 70.21% for white rice, and from 54.19% to 56.14% for red rice. However, Muntana and Srihanam [39] reported lower antioxidant activity for white rice compared to red and black rice. Although phenolic substances are responsible for the antioxidant activity of plant materials, antioxidant potency is not solely characterized by the TPC [40]. Generally, greater antioxidant activity was positively correlated with the TPC in extracts [41]. However, in this study, the phenolic compounds were found at very low content, and thus the free radical scavenging capability varied very slightly. In this investigation, no link was found between TPC inhibition and DPPH• inhibition. Brown rice flour extract with a higher TPC had a lower DPPH• inhibition. Sompong et al. [42] also reported a negative correlation between TPC and DPPH• scavenging activity. Phenolics are not the only determinant of the antioxidant power of plant materials but other phytochemicals with antioxidant activity (e.g. tocopherols, tocotrienals, γ-oryzanol, phytic acid, and carotenoids) can also be included [32, 43, 44]. Those phytochemicals can be co-extracted with the solvent used for TPC extraction and can definitely be found in the final extract [32, 33, 44].

For the FTIR spectra, all the samples showed similar spectra with different peak intensities indicating the same functional groups with different contents were found. Flores-Morales et al. [45] reported that the bands at 400 cm$^{-1}$ and 700 cm$^{-1}$ were associated with the structural vibration of amylose and amylopectin. In addition, the same functional groups were observed in rice flours, including–OH group (3,298–3,278 cm$^{-1}$),–C-H stretch (2,924–2,854 cm$^{-1}$),–

C = O group (1,710–1,000 cm$^{-1}$), mostly aldehyde group, glucose, cyclodextrin, and–C-OH bending vibrations (856–573 cm$^{-1}$). The bands at 3,000–3,500 cm$^{-1}$ corresponded to the O-H stretching vibration regions [46]. The results were in agreement with the earlier reports of Falade and Christopher [31] and Anugrahati et al. [47]. The peak at 1,047 cm$^{-1}$ represented the ordered structure of starch and the peak at 1,022 cm$^{-1}$ was referred to the amorphous structure of starch [48]. Thus, the degree of order in the starch can be estimated from the ratio of peaks at 1,047 cm$^{-1}$/1,022 cm$^{-1}$ [48]. From the calculation, the ratio 1,047 cm$^{-1}$/1,022 cm$^{-1}$ of all samples was fallen in the narrow range (1.24 for CMRF, 1.19 for JMRF, 1.16 for NKRF, and 1.14 for KMRF). Results indicated that the arrangement of starch of all the rice flours in this study was in the ordered structure. The use of FIIR analysis for the detection of rice globulin secondary structure is based on the amide I region composed of C = O stretching vibrations in the region of 1,611–1,690 cm$^{-1}$ [13]. From the results, CMRF had the lowest intensity in this region, followed by KMRF, and JMRF/NKRF, which was in agreement with the protein contents in the flours.

The color of the rice flours were different. The highest $L^*$ value, which express the lightness, was found in CMRF because the CMRF was prepared from the well-milled rice. CMRFs are generally white in color, hence the $L^*$ value of 93.10 reported for CMRF by Rosniyana et al. [23] was not surprising. The obvious differences in rice flour color were affected by their polyphenols which related to the color of the seed [10]. Since the milling and polishing techniques were used in the CMRF manufacture. As a result, all brown rice flours had greater $a^*$ and $b^*$ values with lower $L^*$ values, whereas CMRF had a negative $a^*$ value, a low $b^*$ value, and the highest $L^*$ value, indicating the darker of the former. Higher $a^*$ and $b^*$ values with lower $L^*$ value in the three brown rice flours from this study were related to their total phenolic contents (Table 1). The predominance of phenolics in rice hulls and bran layers led to the increased $a^*$ and $b^*$ of the brown rice [17, 49]. It has been reported that rice with light brown pericarp color presented mainly low molecular weight phenolics whereas in those with dark pericarp color contained the compounds with higher molecular weight [50]. A negative correlation between TPC and $L^*$ value was reported in various non-pigmented and pigmented rice flour samples [5].

The rice flours had bulk densities ranging between 1.36 g/cm$^3$ to 1.83 g/cm$^3$. The bulk density range of 0.65 g/cm$^3$ to 0.89 g/cm$^3$ obtained from some rice flours in Nigeria by Falade and Christopher [31]. The bulk density of flour is generally affected by the composition and particle size. The variation in starch content may have caused a modest fluctuation in bulk density. Iwe et al. [51] suggested that starch content increased the bulk density of flours. The higher carbohydrate content of CMRF may account for its larger bulk density when compared to the others, as shown in Table 1. According to Appiah et al. [15], the higher the carbohydrate content the greater the bulk density. Therefore, the highest bulk density of CMRF/NKRF suggested their suitability be used as a thickening agent in food products.

Other than nutritional aspects, functional features of food are those that are essential for successfully utilizing the food source [52]. The composition and structure of food components, as well as their interactions, influence functional properties [52]. WAC represents the ability of a product to interact with water. It is useful to increase yield and consistency and offer body to the food [53]. WAC of the rice flours ranged from 0.71 to 1.17 g/g. Rosniyana et al. [23] reported the WAC of 0.88 g/g on CMRF. The highest WAC of KMRF and JMRF could be due to the presence of a higher amount of fiber and protein content in these flours. The result implied that KMRF and JMRF would yield thicker pastes when mixed with water. Flours with high WAC could contain more hydrophilic proteins [15]. It has been reported that WAC is an important functional attribute considered in the development of cereal based food since high WAC might improve product cohesiveness [54]. The variation in WAC between flour may be

attributed to different protein concentrations, their degree of association with water, and their conformational aspects. Baxter et al. [55] suggested that the presence of albumin in rice starch facilitated the uptake of water by starch in during cooking due to the formation of protein-water-starch interactions. The presence of globulin initially enhanced the rate of water absorption by rice starch during cooking, but the presence of glutelin slowed it down, according to Baxter et al. [56].

OAC represents the physical entrapment of oil [3]. The OAC of the rice flours in this study ranged from 1.39 g/g to 2.49 g/g. Comparatively, the OAC was higher than reported on FARO 44 rice (OAC = 0.46 g/g) by Iwe et al. [45]. Kraithong et al. [10] reported an OAC range of 1.11–1.34 g/g for some Thai organic rice flours. OAC is a prime functionality that intensifies the mouthfeel while maintaining the flavor of food products. High OAC of the flour suggests its usefulness in lipid-containing foods e.g. bakery products. The higher OAC recorded in the rice flours may be due to the presence of more hydrophobic proteins with superior lipid binding efficacy [28]. Albumin (water-soluble), globulin (salt-soluble), glutelin (alkali-soluble), and prolamin (alcohol-soluble) are the four components of rice protein [57]. The two primary proteins are glutelin (about 80%) and globulin (about 12%), while albumin (about 5%) and prolamin (about 3%) are minor ones [57]. Both glutelin and prolamin are hydrophobic proteins that accumulate in small vacuoles or protein bodies [58]. As a result, rice had a hydrophobic protein level of more than 80%. Due to the similar OAC, any of the rice flours could be chosen and employed in food formulations requiring OAC, such as soups, cakes, and sausages [3].

The differences in swelling power and solubility may be due to the amount of protein, amylose content, and lipids [59]. The current findings were consistent with previous research, which reported swelling power values in rice flours of several rice varieties ranging from 4.7 g/g to 16.23 g/g [18, 60]. More specifically, the swelling power of the flours in this case was similar to Jamal et al. [6] who reported swelling power of 5.74–7.64 g/g for rice flours. For water solubility of rice flour, Kraithong et al. [10] reported the ranges of 2.97–7.05% for some Thai organic rice flours. A lower swelling power and solubility in CMRF can be attributed to a higher degree of intermolecular association and higher amylose content compared to the other rice flours. Swelling is facilitated by amylopectin and disturbed by amylose [61]. The swelling factor of starches is decreased upon the increase of amylose content in starches.

When flour is mixed with water and heated, the LGC represents the lowest amount of flour required to gel. Thus, LGC is an important index of gel-forming ability. Here, the LGC values ranged from 4.0–8.0%. The LGC of the flours was comparable to that of rice flour (6%) reported by Chandra [62]. However, higher LGC values were reported in some rice flours, ranging from 8.0% to 22.0% [3, 17, 63]. A lower LGC suggests stronger gelling capacity in general. The result suggested that KMRF was easier to form gel than JMRF, and NKRF/CMRF, respectively. The presence of varying levels of protein, carbohydrates, and lipids in the rice varieties can cause the differences in LGC [63]. The FC of the flours in this study (1.30–2.60%) was lower than the previous reports. The FC of the rice flour of 3.52% has been reported by Chandra [62]. However, higher foaming capacity of 10.40% has been reported for polished FARO 44 rice variety flour [51]. This might be due to its relatively high protein content (16%) of the flour form polished FARO 44 rice variety. The brown rice varieties with higher protein contents (Table 1) had slightly higher foam capacity than the CMRF. A lower protein content in the rice flours from this experiment may have resulted in the general lower FC of the flours. The FC of flour is dependent on the protein content because proteins are surface-active agents which can create the film at the interphase to trap the air bubbles [17, 62]. The FC seemed to be related to the solubility, in which the lowest value was found in CMRF (Table 2). The foam capacity of a food material is governed by several factors such as the type and concentration of

protein, solubility, and other parameters [64]. The low FC was desirable for flours intended for the application of some bakery products such as crackers and biscuits [17, 62].

Flours are commonly mixed with water for proper hydration before being subjected to thermal treatments that can cause a variety of physicochemical changes in the various components of flours, including starch gelatinization, protein denaturation, enzyme inactivation, and interactions between them [65]. Flours can serve as a thickening agent, gelling agent, binder, and/or stabilizer in final food items as a result of various physicochemical processes. Some of the flour pastes can form a gel after cooling and storage, which is a viscoelastic solid-in-liquid colloid with a definite structure but no fluidity [66]. Pasting property of flours is one of the important functional characteristics that govern how they are used in food products. The pasting profiles of flour are species-dependent and are influenced by the chemical composition of flour [67]. The molecular interactions of starch with other components such as proteins, lipids and non-starch polysaccharides, are some of the factors involved in determining cooking behavior of rice [22]. Brown rice flours had a higher pasting temperature than CMRF ($p < 0.05$). The CMRF was made from white rice, which was easier to absorb water, swell, and gelatinize [68]. Thus, the temperature needed for the gelatinization of CMRF was lower than the others. Cooking the brown rice flours would, therefore, require more energy than the CMRF. These data back with the theory that rice varieties with higher amylose content require less time to cook. Riceberry, a pigmented rice flour, had significantly higher pasting temperature (87.8˚C) than Phitsanulok (white rice flour) (86.8˚C) as reported by Kraithong et al. [10]. White rice flours have lower pasting temperatures than brown rice flours, according to research findings. This could be due to poor interactions between starch granules, which results in the lowest pasting temperature. When a starch suspension was subjected to heat above a critical temperature, granules underwent swelling and amylose dripped into the aqueous phase, resulting in increased viscosity [69]. The creation of a starch-lipid complex in brown rice varieties prevents water from reaching the starch granules, necessitating a higher temperature because the starch granules have stronger bonds.

The peak viscosity of the rice flours in this study was ranged from 847 BU to 2,250 BU. However, a higher peak viscosity values of 2,376 to 3,988 BU was reported for six Nigerian rice flours [31]. Peak viscosity is the indicator of starch granule swelling and high-value peak viscosity indicates a high capacity of swelling of starch [69]. Starches with high peak viscosity are possible to show high breakdown values, leading to weak gels. Such gels are prone to be destroyed under shear and heat. This means that peak viscosity and break down have a positive relationship. The higher peak viscosity in CMRF was due to a larger breakdown during heating due to weaker heat and shear stress resistance [55]. As a result, the fact that CMRF had the greatest breakdown value of 321 BU was unsurprising. The high peak viscosity in the CMRF may be because of its initial high amylose (starch content). The low peak viscosity in the brown rice flours may be due to their initial high fat and protein contents. Swelling power is retarded by the presence of fat and protein. Flours with a high starch content have a high gelatinization and swelling index. According to the findings, the CMRF and JMRF could be useful thickening agents. As stated by Kraithong et al. [10], higher peak viscosity (4,067 BU) was observed in brown jasmine rice flour and the lowest value (2,260 BU) was from white rice (Phitsanulok). Rice flour with high viscosity is commonly used as a thickening agent in high viscosity food [28]. Trough viscosity is the viscosity reaching the minimum after cooling [51]. CMRF and JMRF with relatively high amylose content had a higher trough viscosity. The same trend was described by Sompong et al. [42] who reported a positive correlation between trough viscosity and amylose content.

The setback is the tendency of flours to reassociate and retrograde on cooling. Setback values were correlated with the gelling ability of starches to form semi-solid pastes. The higher

setback values for the JMRF suggested that it had a greater chance to retrograde than the other rice flours. Flours with resistant to retrograde are suitable for soups and sauces because they can prevent the loss of viscosity and precipitation [69]. A high setback is related with the syneresis. Thus, the lower setback values in NKRF and CMRF indicated that it would form a better flour paste that could find applications in the confectionery industries. Breakdown viscosity is indicative of paste stability [31]. The higher the breakdown viscosity, the lower the ability of the starch to resist shear stress and heating [31]. Thus, JMRF and KMRF, with the lowest breakdown values might be able to withstand more heating and shear stress than others.

The final viscosity is the tendency of flours to form paste or gel after cooling. Several studies showed an increase in final peak viscosity than their corresponding peak viscosity. The same trend was observed in this study for JMRF and KMRF but not for NKRF and CMRF. Other chemical components (e.g. lipid and protein contents) and swelling power may all have a part in flour gelatinization, explaining the observed trend. In general, peak, trough, and breakdown viscosity of rice flour were positively correlated with amylose content [28]. Thus, the differences in final viscosity values among the rice flours may be due to the differences in amylose content and other chemical constituents, particularly the type and amount of proteins. Baxter et al. [56] found that the presence of glutelin in rice starch caused an increase in pasting temperature but a decrease in the viscosity parameters of the starch paste. Contrastingly, the presence of globulin resulted in a decrease in all the pasting parameters. In the case of prolamin, the presence of prolamin in rice starch facilitated the water absorption during cooking but the gelatinized starch absorbed less water compared with control samples without prolamin [70].

## Conclusion

This study revealed the variations in chemical, physical, functional, and pasting properties that exist among four rice flours in Thailand. The compositional variations greatly influenced the techno-functionality and final quality of rice flours. The brown rice flours in this study were classified as a low amylose type whereas the CMRF was an intermediate type. All rice flours had low total phenolic content with moderate DPPH$^\bullet$ scavenging activity. Functionalities and pasting properties of the flours varied among the cultivars. Those variations may be due to the differences in genetic constitution and processing condition. In addition to their nutritional and technological benefits, local rice flours have been revealed to be a potential source of bioactive secondary metabolites, which might be used as functional food ingredients in both domestic and industrial applications.

## Acknowledgments

We would like to thank Food Technology and Innovation Center of Excellence, Walailak University for providing the scientific and technological equipment for this research.

## Author Contributions

**Conceptualization:** David Oppong, Worawan Panpipat, Manat Chaijan.

**Data curation:** David Oppong.

**Formal analysis:** Worawan Panpipat.

**Funding acquisition:** Manat Chaijan.

**Investigation:** David Oppong.

**Methodology:** David Oppong, Worawan Panpipat, Manat Chaijan.

**Resources:** Worawan Panpipat, Manat Chaijan.

**Supervision:** Worawan Panpipat, Manat Chaijan.

**Validation:** Manat Chaijan.

**Writing – original draft:** David Oppong, Worawan Panpipat, Manat Chaijan.

**Writing – review & editing:** David Oppong, Worawan Panpipat, Manat Chaijan.

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
