## [Decision Letter · Decision Letter 0]

30 Jun 2021

PONE-D-21-17663

Chemical, physical, and functional properties of Thai indigenous brown rice flour

PLOS ONE

Dear Dr. Chaijan,

Thank you for submitting your manuscript to PLOS ONE. After careful consideration, we feel that it has merit but does not fully meet PLOS ONE’s publication criteria as it currently stands. Therefore, we invite you to submit a revised version of the manuscript that addresses the points raised during the review process.

We look forward to receiving your revised manuscript.

Kind regards,

Umakanta Sarker

Academic Editor

PLOS ONE

Journal Requirements:

Reviewers' comments:

Reviewer's Responses to Questions

**Comments to the Author**

1. Is the manuscript technically sound, and do the data support the conclusions?

Reviewer #1: Yes

Reviewer #2: Yes

2. Has the statistical analysis been performed appropriately and rigorously? 

Reviewer #1: Yes

Reviewer #2: Yes

3. Have the authors made all data underlying the findings in their manuscript fully available?

Reviewer #1: Yes

Reviewer #2: Yes

4. Is the manuscript presented in an intelligible fashion and written in standard English?

Reviewer #1: Yes

Reviewer #2: Yes

5. Review Comments to the Author

Reviewer #1: Comments to the Author

The manuscript is focused on the chemical, physical, and functional properties of Thai indigenous brown rice flour in comparison with JMRF) and CMRF. Overall, the manuscript is well designed and well written. However, the manuscript needs further improvement before acceptance. The authors should pay attention to the following specific points.

Specific comments:

1. The Abstract should be a bit more concise and needs further improvement.

2. Thoroughly check the manuscript for grammatical and syntax errors. Some sentences need to be paraphrased with proper scientific wordings.

3. Line 71: Change “which is” to “which are”

4. Line 90: Change “gain” to “get”

5. Lines 102,103: The sentence needs improvement

6. Lines 146, 147: The two sentences should be merged together. Write it as “….with gentle stirring and then centrifuged at …..”

7. Lines 181, 182: The sentence needs to be improved.

8. Line 186: “were reported”?. It is not very clear and need to be paraphrased properly. Such as “…. Setback were measured according to the method of…”

9. Line 188: Should be “…. And the experiments were performed in triplicate”.

10. The subtitles of the Results section should be written based on the results of your study.

11. Line 196: Add comma before “whereas”

12. Line 200: Remove comma after “KFRF/JMRF,”

13. Line 269: Add comma before “as shown in Table 3.”

14. Line 271: Add comma before “as illustrated in Table 3.”

15. No need to give subtopics in the Discussion section

16. Lines 290, 291: The sentence needs to be paraphrased.

17. The discussion section is too long and needs to be properly revised and should be written a bit brief. Moreover, there are some minor syntax and grammatical errors, which should be carefully checked.

18. In the conclusion, write about the significance of your study. What benefits it could provide to the industry.

Reviewer #2: 1. In title add flours

2. Line 30, Commercial flour is from which variety?? What are the specialties for this??

3. Key words maintain alphabetically

4. Line 56, commercial rice flour prepared with what variety??

5. Line 62-63, better mention all factors

6. Use of abbreviations are improper. Give abbreviation initially, than follow giving abbreviations (line number 74)

7. Line 84, is it true altitude is only 6 meters?? Cross check once.

8. Line 89, 90. How much time takes to determine the analysis?? at room temperature there may be many changes takes place in composition. for example Lipid degradation (Give clarity here)

9. What is the conversion factor used for determination of protein? (Line 93)

10. Line 139, replace poured in to taken

11. Line 174, given centrifuge as RPM however, before given as G. Better use uniform units in document

12. Statistical analysis given completely randomization, As your design is a single factor how did this??

13. In total discussion given dw, Better mention initially once, than can avoid every place as (dw)

14. In each table give P vales and CV values.

15. Line 213, sure the means are comparing in row wise or column wise?? Check once, You have to compare the column wise. Because your factor is rice type not the character of the rice. Also, correct the discussion accordingly.

16. Comment 15 should consider for all the tables.

17. Line 223, How considered Moderate?

18. Line 278, The discussion started and the experiment considered is not matching. As your work not have any geographical variation

20. Line 287, What about the protein content with other rice varieties??

21. Line 291, what values?

22. Line 292, what are this some selected varieties? They are indigenous??

23. Better discuss implication of the composition on the technical, consumption quality. For example, issues related to fiber on technical quality

24. Line 3050306. How brown rice has low fat than the white rice? Need justifications

25. Line 315-316. There is a lot of variation with your study.. What may be the reason??

26. Line 325-326. The bran extract and total rice flour is different.. Better to compare with the suitable rice flours study

27. Line 330. Can give a reference that how much removed by milling and how much degraded by flour preparations??

28. You are not done any correlation, but discussed many places, as correlated. If you done correlation, place the table and discuss with r and p values.

29. Line 441. Check space before full stop

30. Line 430-431, how much hydrophobic protein present in rice??

31. Line 439, for what solubility??

32. Line 465, The pasting should discuss with the applications and benefits.

33. Section conclusions should be given for the recommendations related to the industrial and food applications

6. PLOS authors have the option to publish the peer review history of their article (what does this mean?). If published, this will include your full peer review and any attached files.

Reviewer #1: No

Reviewer #2: **Yes: **Dr. Neela Satheesh

---

## [Author Response · Author response to Decision Letter 0]

3 Jul 2021

Response to Reviewers

All points raised by the reviewers were carefully addressed and answered point-by-point. A revision was made in highlighted red fonts. The revised manuscript was carefully rechecked for English using paraphrasing tool. 

Journal Requirements:

 Ans: The revised manuscript was carefully prepared to meet PLOS ONE's style requirements.

Reviewer #1: Comments to the Author

The manuscript is focused on the chemical, physical, and functional properties of Thai indigenous brown rice flour in comparison with JMRF) and CMRF. Overall, the manuscript is well designed and well written. However, the manuscript needs further improvement before acceptance. The authors should pay attention to the following specific points.

Specific comments:

1. The Abstract should be a bit more concise and needs further improvement.

Ans: The abstract was updated and made shorter.

2. Thoroughly check the manuscript for grammatical and syntax errors. Some sentences need to be paraphrased with proper scientific wordings.

Ans: The revised manuscript was carefully rechecked for English using paraphrasing tool. 

3. Line 71: Change “which is” to “which are”

Ans: Done.

4. Line 90: Change “gain” to “get”

Ans: Done.

5. Lines 102,103: The sentence needs improvement

Ans: It was changed to “A calibration curve was created using standard amylose obtained from potato starch to quantify the amylose content.”

6. Lines 146, 147: The two sentences should be merged together. Write it as “….with gentle stirring and then centrifuged at …..”

Ans: It was merged accordingly. “The suspensions were heated in a water bath at 30 °C for 30 min with gentle stirring and then centrifuged at 1,500 �g for 10 min (RC-5B plus centrifuge).”

7. Lines 181, 182: The sentence needs to be improved.

Ans: It was changed to “In a canister, the rice flour sample (3 g) was inserted, and then 25 mL of distilled water was added (14% moisture basis).”

8. Line 186: “were reported”?. It is not very clear and need to be paraphrased properly. Such as “…. Setback were measured according to the method of…”

Ans: It was changed to “The pasting parameters e.g. pasting temperature, peak viscosity, breakdown, final viscosity, and setback were measured according to the method of Kraithong et al. [10].”

9. Line 188: Should be “…. And the experiments were performed in triplicate”.

Ans: It was changed to “A completely randomized design was used and the experiments were performed in triplicate.”

10. The subtitles of the Results section should be written based on the results of your study.

Ans: Done.

11. Line 196: Add comma before “whereas”

Ans: Done.

12. Line 200: Remove comma after “KFRF/JMRF,”

Ans: Done.

13. Line 269: Add comma before “as shown in Table 3.”

Ans: Done.

14. Line 271: Add comma before “as illustrated in Table 3.”

Ans: Done.

15. No need to give subtopics in the Discussion section

Ans: The subtopics in the Discussion section were removed as suggested.

16. Lines 290, 291: The sentence needs to be paraphrased.

Ans:

17. The discussion section is too long and needs to be properly revised and should be written a bit brief. Moreover, there are some minor syntax and grammatical errors, which should be carefully checked.

Ans: The discussion section was revised using the paraphrasing tool.

18. In the conclusion, write about the significance of your study. What benefits it could provide to the industry.

Ans: The industrial and food applications were added. “In addition to their nutritional and technological benefits, local rice flours have been revealed to be a potential source of bioactive secondary metabolites, which might be used as functional food ingredients in both domestic and industrial applications”

Reviewer #2: 

1. In title add flours

Ans: Done.

2. Line 30, Commercial flour is from which variety?? What are the specialties for this??

Ans: It was Chai Nat 1 variety which was a non-glutinous white rice. So, it was specified in the Abstract and in the Material and methods.

3. Key words maintain alphabetically

Ans: Done.

4. Line 56, commercial rice flour prepared with what variety??

Ans: Chai Nat 1 variety was added.

5. Line 62-63, better mention all factors

Ans: It was changed to “Compositional variations in terms of physical, chemical, thermal, and pasting properties were found among rice varieties, depending on the genotype, agronomic and cultivation conditions, environmental factors, storage conditions, and processing parameters [4, 6].”

6. Use of abbreviations are improper. Give abbreviation initially, than follow giving abbreviations (line number 74)

Ans: CMFR was first mentioned in the first paragraph of Introduction. So, the full name was not applied for the second use.

7. Line 84, is it true altitude is only 6 meters?? Cross check once.

Ans: It was due to the mistyping. Actually, it is 9 meters.

8. Line 89, 90. How much time takes to determine the analysis?? at room temperature there may be many changes takes place in composition. for example Lipid degradation (Give clarity here)

Ans: Thank you very much for your suggestion. The storage time was less than a month.

9. What is the conversion factor used for determination of protein? (Line 93)

Ans: A conversion factor of 5.95 was used. It was stated in the text already.

10. Line 139, replace poured in to taken

Ans: Done.

11. Line 174, given centrifuge as RPM however, before given as G. Better use uniform units in document

Ans: In Line 174, it was not a centrifugation force but it was a homogenization speed. So, we would like to keep “rpm” for this case.

12. Statistical analysis given completely randomization, As your design is a single factor how did this??

Ans: A completely randomized design was used for experimental design with a single factor of four rice flours and the experiments were performed in triplicate. Data were subjected to one-way analysis of variance (ANOVA). Duncan’s multiple-range test was used to analyze significant differences (p<0.05) among samples, using the SPSS program (Version 23.0, SPSS Inc., Chicago, IL, USA).

13. In total discussion given dw, Better mention initially once, than can avoid every place as (dw)

Ans: The sentence was added in the Material and methods to indicate that “The contents for all chemical compositions were reported on a dry weight (dw) basis.” As a result, except in the Table, the phrase "dw" was removed from the rest of the text.

14. In each table give P vales and CV values.

Ans: In all Table in this study, values are means ± standard deviation from triplicate determinations. Different letters are significantly different along the rows (p<0.05). The different letters were used to signify significant differences at p<0.05. This was the standard format for scientific papers. The CV values were excluded from the report.

15. Line 213, sure the means are comparing in row wise or column wise?? Check once, You have to compare the column wise. Because your factor is rice type not the character of the rice. Also, correct the discussion accordingly.

Ans: Because rice flour is an independent variable and chemical composition is a dependent variable, it was sure that the means were compared row by row. The goal of the experiment was to investigate if the chemical composition/properties of rice flour differed depending on the variety.

16. Comment 15 should consider for all the tables.

Ans: We checked all the Table already and we confirmed the answer in Comment 15.

17. Line 223, How considered Moderate?

Ans: To make the sentence clearer, it was changed. “Inactivation of DPPH● was found in all of the rice flour extracts examined, ranging from 38.87 to 46.77%.”

18. Line 278, The discussion started and the experiment considered is not matching. As your work not have any geographical variation

Ans: The commercial rice flour (CMRF) was made from Chai Nat 1 rice farmed in the Central part of Thailand, whereas all three brown rice flours were made in the Southern part of Thailand, as stated in the "Raw materials" section. As a result, the geographical variation can be identified.

20. Line 287, What about the protein content with other rice varieties??

Ans: We originally stated in Line 301-303 that “The protein amount found in this study was comparable to that found in Thai purple rice flours (6.6-13.0%) [18], black glutinous rice flour (8.0%) [20], and black rice variety (8.0%) [21].”

21. Line 291, what values?

Ans: 1.99-2.22%.

22. Line 292, what are this some selected varieties? They are indigenous??

Ans: Some of them are indigenous. So, the sentence was revised and the detail was given. “The results of this study were within the range of 0.43-2.34% reported by Devi et al. [19] on 92 rice varieties, including indigenous, improved, and fragrant types.”

23. Better discuss implication of the composition on the technical, consumption quality. For example, issues related to fiber on technical quality

Ans: A revision was made as suggested.

24. Line 305-306. How brown rice has low fat than the white rice? Need justifications

Ans: We stated that “The fat content of CMRF in this research was lower than Phitsanulok white rice (1.13%) [10].”. CMRF stands for commercial rice flour made from Chai Nat 1 white rice rather than brown rice. Both are white rice in this situation.

25. Line 315-316. There is a lot of variation with your study.. What may be the reason??

Ans: Rice variety was an independent variable in this study, while chemical composition was a dependent variable. As a result, rice variety largely determined the different amylose content.

26. Line 325-326. The bran extract and total rice flour is different.. Better to compare with the suitable rice flours study

Ans: New reference on rice flour was used instead. 

27. Line 330. Can give a reference that how much removed by milling and how much degraded by flour preparations??

Ans: The reference was given. “Grain phenolic compounds were eliminated during rice milling and flour preparation, as indicated by a low TPC in CMRF. In comparison to unmilled rice, Sapna et al. [38] found that the TPC of milled rice retained roughly 73-89% of the overall TPC.”

28. You are not done any correlation, but discussed many places, as correlated. If you done correlation, place the table and discuss with r and p values.

Ans: The statement was changed to “In this investigation, no link was found between TPC inhibition and DPPH● inhibition.” because the correlation was not done in this study.

29. Line 441. Check space before full stop

Ans: Done.

30. Line 430-431, how much hydrophobic protein present in rice??

Ans: Albumin (water-soluble), globulin (salt-soluble), glutelin (alkali-soluble), and prolamin (alcohol-soluble) are the four components of rice protein [58]. The two primary proteins are glutelin (about 80%) and globulin (about 12%), while albumin (about 5%) and prolamin (about 3%) are minor ones [58]. Both glutelin and prolamin are hydrophobic proteins that accumulate in small vacuoles or protein bodies [59]. As a result, rice had a hydrophobic protein level of more than 80%. 

31. Line 439, for what solubility??

Ans: It was changed to “For water solubility of rice flour,…”

32. Line 465, The pasting should discuss with the applications and benefits.

Ans: A revision was made as suggested. References were added.

33. Section conclusions should be given for the recommendations related to the industrial and food applications

Ans: The industrial and food applications were added. “In addition to their nutritional and technological benefits, local rice flours have been revealed to be a potential source of bioactive secondary metabolites, which might be used as functional food ingredients in both domestic and industrial applications”

---

## [Decision Letter · Decision Letter 1]

22 Jul 2021

Chemical, physical, and functional properties of Thai indigenous brown rice flours

PONE-D-21-17663R1

Dear Dr. Chaijan,

We’re pleased to inform you that your manuscript has been judged scientifically suitable for publication and will be formally accepted for publication once it meets all outstanding technical requirements.

Kind regards,

Umakanta Sarker

Academic Editor

PLOS ONE

Additional Editor Comments (optional):

Reviewers' comments:

Reviewer's Responses to Questions

**Comments to the Author**

1. If the authors have adequately addressed your comments raised in a previous round of review and you feel that this manuscript is now acceptable for publication, you may indicate that here to bypass the “Comments to the Author” section, enter your conflict of interest statement in the “Confidential to Editor” section, and submit your "Accept" recommendation.

Reviewer #1: All comments have been addressed

Reviewer #2: All comments have been addressed

2. Is the manuscript technically sound, and do the data support the conclusions?

Reviewer #1: Yes

Reviewer #2: Yes

3. Has the statistical analysis been performed appropriately and rigorously? 

Reviewer #1: Yes

Reviewer #2: Yes

4. Have the authors made all data underlying the findings in their manuscript fully available?

Reviewer #1: Yes

Reviewer #2: Yes

5. Is the manuscript presented in an intelligible fashion and written in standard English?

Reviewer #1: Yes

Reviewer #2: Yes

6. Review Comments to the Author

Reviewer #1: The authors have modified the paper according to the reviewers suggestion. It should be accepted now.

Reviewer #2: Author are addressed all the comments however, the following two comments has to check

1. I am still suggesting to add P-vales to the tables. P-vale showed the how much significant.

7. PLOS authors have the option to publish the peer review history of their article (what does this mean?). If published, this will include your full peer review and any attached files.

Reviewer #1: **Yes: **Ishfaq Ahmed

Reviewer #2: **Yes: **Dr. Neela Satheesh

---

## [Editor Report · Acceptance letter]

26 Jul 2021

PONE-D-21-17663R1 

Chemical, physical, and functional properties of Thai indigenous brown rice flours 

Dear Dr. Chaijan:

I'm pleased to inform you that your manuscript has been deemed suitable for publication in PLOS ONE. Congratulations! Your manuscript is now with our production department. 

Kind regards, 

on behalf of

Professor Umakanta Sarker 

Academic Editor

PLOS ONE